# Bioactive Sesquiterpenoids from *Santolina chamaecyparissus* L. Flowers: Chemical Profiling and Antifungal Activity Against *Neocosmospora* Species

**DOI:** 10.3390/plants14020235

**Published:** 2025-01-16

**Authors:** Eva Sánchez-Hernández, Jesús Martín-Gil, Vicente González-García, José Casanova-Gascón, Pablo Martín-Ramos

**Affiliations:** 1Department of Agricultural and Forestry Engineering, ETSIIAA, Universidad de Valladolid, Avenida de Madrid 44, 34004 Palencia, Spain; eva.sanchez.hernandez@uva.es (E.S.-H.); jesus.martin.gil@uva.es (J.M.-G.); 2Departamento de Sistemas Agrícolas, Forestales y Medio Ambiente, Centro de Investigación y Tecnología Agroalimentaria de Aragón (CITA), Avda. Montañana 930, 50059 Zaragoza, Spain; vgonzalezg@aragon.es; 3Instituto Agroalimentario de Aragón-IA2 (Universidad de Zaragoza-CITA), Avda. Montañana 930, 50059 Zaragoza, Spain; 4Instituto Agroalimentario de Aragón-IA2 (Universidad de Zaragoza-CITA), EPS, University of Zaragoza, Carretera de Cuarte s/n, 22071 Huesca, Spain; jcasan@unizar.es

**Keywords:** aromadendrane skeleton, biorational pesticides, cotton-lavender, hydroazulene framework, tomato, zucchini

## Abstract

*Santolina chamaecyparissus* L. (cotton-lavender) is receiving increasing attention due to its potential for modern medicine and is considered both a functional food and nutraceutical. In this work, the phytochemical profile of its flower hydromethanolic extract was investigated by gas chromatography–mass spectrometry, and its applications as a biorational for crop protection were explored against *Neocosmospora* spp., both in vitro and in planta. The phytochemical profiling analysis identified several terpene groups. Among sesquiterpenoids, which constituted the major fraction (50.4%), compounds featuring cedrane skeleton (8-cedren-13-ol), aromadendrene skeleton (such as (−)-spathulenol, ledol, alloaromadendrene oxide, epiglobulol, and alloaromadendrene), hydroazulene skeleton (ledene oxide, isoledene, and 1,2,3,3a,8,8a-hexahydro-2,2,8-trimethyl-,(3a*α*,8*β*,8a*α*)-5,6-azulenedimethanol), or copaane skeleton (*cis*-α-copaene-8-ol) were predominant. Additional sesquiterpenoids included longiborneol and longifolene. The monoterpenoid fraction (1.51%) was represented by eucalyptol, (+)-4-carene, endoborneol, and 7-norbornenol. In vitro tests against *N. falciformis* and *N. keratoplastica*, two emerging soil phytopathogens, resulted in effective concentration EC_90_ values of 984.4 and 728.6 μg·mL^−1^, respectively. A higher dose (3000 μg·mL^−1^) was nonetheless required to achieve full protection in the in planta tests conducted on zucchini (*Cucurbita pepo* L.) cv. ‘Diamant F1’ and tomato (*Solanum lycopersicum* L.) cv. ‘Optima F1’ plants inoculated with *N. falciformis* by root dipping. The reported data indicate an antimicrobial activity comparable to that of fosetyl-Al and higher than that of azoxystrobin conventional fungicides, thus making the flower extract a promising bioactive product for organic farming and expanding *S. chamaecyparissus* potential applications.

## 1. Introduction

Santolina (*Santolina chamaecyparissus* L., family *Asteraceae*), sometimes called cotton-lavender, is an herbaceous perennial shrub native to the Mediterranean region and parts of Europe and America [1] (Figure 1a). It has a mounded, round, and dense habit, reaching only 0.5 m in height and width, with vibrant yellow, button-like composite flowerheads (consisting only of disc florets) perched on stems above the foliage (Figure 1b). It has either silvery-grey or green leaves reminiscent of conifers. When bruised, Santolina foliage emits a pungent aroma reminiscent of a mixture of camphor and resin.

*Santolina chamaecyparissus*’ phytoconstituents have recognized medicinal properties, with analgesic, anticancer, anti-inflammatory, antimicrobial, antioxidant, and antispasmodic activity, as well as central nervous system depressants [3].

Terpenes are the largest and most diverse plant secondary metabolites in nature, serving as informative and defensive vehicles used by plants for antagonistic and mutualistic interactions. Their biosynthesis occurs through two independent pathways: the mevalonate pathway (MVA) in the cytosols of plants and the 2C-methyl-D-erythritol-4-phosphate (MEP) pathway in plastids. Both pathways lead to the formation of isopentenyl diphosphate (IPP) and its isomer dimethylallyl diphosphate (DMAPP), which serve as the precursors for all terpene classes. Monoterpenes are highly diverse compounds occurring in monocotyledonous and dicotyledonous angiosperms, fungi, bacteria, and gymnosperms. They are odoriferous compounds that partly account for the scent of many flowers and fruits and are vital components of essential oils that give plants aroma and flavor. Sesquiterpenes, which share ring classification with monoterpenes (with the exception of a few tricyclic terpenes), demonstrate significant structural diversity arising from the arrangement of their 15-carbon skeletons, the layering of functional groups, and the substituents on their backbone. Both monoterpenes and sesquiterpenes play crucial roles in plant defense against biotic stresses such as pathogenic microbes and herbivore pests [4].

Phytochemical analyses have revealed that cotton-lavender contains various mono- and sesquiterpenoids alongside other secondary metabolites [5]. The bioactive properties of these terpenes vary depending on their composition and relative concentrations, which are influenced by plant origin, plant part used, and extraction method. For instance, a 1,8-cineole- and *β*-eudesmol-rich flowerhead essential oil demonstrated potent antibacterial properties against *Pseudomonas aeruginosa* (Schroeter) Migula and *Enterococcus faecalis* (Andrewes and Horder) Schleifer and Kilpper-Balz, with minimum inhibitory concentration (MIC) values as low as 0.625 μg·mL^−1^ [6].

Although previous studies have investigated the antimicrobial activity of *S. chamaecyparissus* essential oils and extracts against various human pathogens [3], their potential as biorational agents for crop protection remains unexplored. In particular, no studies have examined the activity of cotton-lavender essential oils or extracts against *Neocosmospora* species. These fungi significantly impact industry, human health, and agriculture [7]. Several species are known pathogens in immunocompromised humans, often resulting in high mortality rates despite antifungal therapy [8], while others serve as opportunistic soil-borne phytopathogens, inducing cankers, stem and root rot, and blight in a variety of plants [9].

This work aimed to investigate the phytoconstituents of the hydromethanolic extract of *S. chamaecyparissus* flowers and to examine its antifungal activity—both in vitro and in planta—against two emerging *Neocosmospora* phytopathogens, namely *Neocosmospora falciformis* (Carrión) Summerb. & Schroers, associated with multiple plant species decline and muskmelon wilting and root rot in Spain [10], and *Neocosmospora keratoplastica* Geiser, O’Donnell, Short & Zhang, which affects cucurbits through rot and root decay [11].

## 2. Results

### 2.1. Vibrational Characterization

The Fourier-transform infrared (FTIR) spectrum of the flower extract (Table 1 and Appendix A) displayed characteristic absorption bands assignable to hydroxyl (3292 cm^−1^) and ester carbonyl (1694 cm^−1^) functional groups. Additional bands were observed at 2927, 1596, 1515, 1371, 1258, 1158, 1115, 1031, 854, and 811 cm^−1^ (aromatics). Comparison of the spectrum with that of shade-dried flower heads before extraction showed similar functional groups with only minor shifts, indicating successful ultrasonication-assisted extraction. The concurrence of bands at 3300, 1515, and 811 cm^−1^ suggests the presence of sesquiterpenes [12].

### 2.2. Gas Chromatography–Mass Spectrometry Characterization

Gas chromatography–mass spectrometry (GC-MS) analysis of the extract led to the identification of multiple compounds (Appendix A). Among sesquiterpenoids, which constituted the major fraction (50.4%), compounds featuring the cedrane skeleton, such as 8-cedren-13-ol (27.43%); the aromadendrane skeleton, such as (−)-spathulenol (3.06%), alloaromadendrene and its oxides (2.72%), ledol (2.22%), epiglobulol (1.45%), and aromadendrene (0.65%); the hydroazulene skeleton, including ledene oxide (6.34%), isoledene (1.41%), and 1,2,3,3a,8,8a-hexahydro-2,2,8-trimethyl-,(3aα,8β,8aα)-5,6-azulenedimethanol (0.44%); and the copaane skeleton, represented by *cis*-α-copaene-8-ol (0.77%), were predominant. Additional sesquiterpenoids included perhydrocyclopropa[e]azulene-4,5,6-triol (1.01%), characterized by a fully saturated (perhydro) bicyclo[5.3.0] system with three hydroxyl groups, and spiro[2,4,5,6,7,7a-hexahydro-2-oxo-4,4,7a-trimethylbenzofuran]-7,2’-(oxirane) (1.81%), which is a modified sesquiterpenoid in which the original C15 skeleton has undergone oxidative modifications. The series was completed by longiborneol (0.69%) and longifolene (0.37%), which feature a bicyclo[2.2.1] and a bicyclo[2.2.2] system, respectively (Figure 2). The monoterpenoid fraction (1.51%) was represented by the cyclic ether eucalyptol, the bicyclic (+)-4-carene, and two compounds with bicyclic[2.2.1] systems: endo-borneol (bornane-type) and 7-norbornenol (norbornane-type). Non-terpenoid compounds were also present, with 2,1,3-benzothiadiazole (12%) being the second most abundant component, followed by acetic acid (3.72%), retinol acetate (2.27%), and catechol (1.62%).

### 2.3. In Vitro Antifungal Activity

Figure 3 and Appendix A show the antifungal susceptibility test results. Statistically significant differences in mycelial growth were observed with increasing concentrations of *S. chamaecyparissus* extract in all cases. In particular, the MICs were 1000 and 1500 μg·mL^−1^ for *N. falciformis* and *N. keratoplastica*, respectively. The 50% and 90% effective concentrations (EC_50_ and EC_90_, respectively) were 393.4 and 984.4 μg·mL^−1^ against *N. falciformis* and 281.0 and 728.6 μg·mL^−1^ against *N. keratoplastica.*

### 2.4. In Planta Activity Against N. falciformis

Given that *N. falciformis* was both highly virulent and the most resistant to the treatment in the in vitro tests, it was chosen for further in planta testing. Protection with the *S. chamaecyparissus* flower extract was assayed both on zucchini (*Cucurbita pepo* L.) cv. ‘Diamant F1’ (Figure 4a–d) and tomato (*Solanum lycopersicum* L.) cv. ‘Optima F1’ (Figure 4e–h) plants. In both cases, positive controls (plants artificially inoculated and treated with bi-distilled water only) exhibited the typical symptoms for this pathogen, including yellowing and wilting of leaves, rotting at the stem base and upper root, and—in the case of zucchini plants—the collapse of the entire plant (Figure 4b). Plants treated with the extract at a concentration equal to the MIC (1500 μg∙mL^−1^) showed disease incidence but without the severity observed in the positive control. Based on plant weight results (Figure 5), zucchini plants were more susceptible than tomato plants, probably because the former is a more specific host for this pathogen. Increasing the treatment concentration to twice the MIC (i.e., 3000 μg∙mL^−1^) fully prevented symptoms of wilt and root rot without causing phytotoxicity, resulting in treated plants that appeared identical to the negative controls and with no significant statistical differences except in the case of the zucchini weight, with a 5% lower median for the MIC × 2-treated plants (3437.5 mg) compared to the negative control (3640 mg).

## 3. Discussion

### 3.1. Phytochemical Profile and the Antimicrobial Activity of the Phytoconstituents

The characterization of the *S. chamaecyparissus* flower hydromethanolic extract revealed a complex composition that was dominated by sesquiterpenoids. According to the comprehensive review by Tundis and Loizzo [14] on the phytochemistry of the genus *Santolina*, previous studies have reported the presence of terpenoids, particularly eudesmane and germacrane sesquiterpenoids, along with chrysanthemane monoterpenoids. Among the sesquiterpenoids identified herein, spathulenol was previously reported in *S. africana* and *S. pectinata* and aromadendrene only in *S. corsica* [14].

The phytochemical profile of *S. chamaecyparissus* flower extract studied herein shows the closest similarity to two previously reported sesquiterpenoid profiles: that of the essential oil obtained from fresh leaves of *Duguetia glabriuscula* (R.E.Fr.) R.E.Fr. [15], containing ledol, spathulenol, alloaromadendrene, alloaromadendrene oxide, epiglobulol, and ledene oxide, and that of the fruit volatiles from *Eucalyptus camaldulensis* Denham [16], which contain ledene, aromadendrene, alloaromadendrene, globulol, and isolongifolen. Although the three species belong to the class Magnoliopsida, they are classified into different orders and families. Therefore, their shared bioactivity should be attributed to the nature of their phytochemical components rather than to their taxonomic proximity.

The most abundant constituent of *S. chamaecyparissus* flower hydromethanolic extract was 8-cedren-13-ol (27.4%) Previous reports have documented its presence in various aromatic and medicinal plants, including *Artemisia arborescens* L. [17], *Curcuma* species [18], *Juniperus recurva* Buch. [19], *Mentha longifolia* L. [20], *Peucedanum longifolium* Waldst. & Kit. [21], *Plumeria alba* L. [22], and *Zingiber officinale* Roscoe. The antimicrobial activity of cedrene and its derivatives are supported by previous research on *Tetraclinis articulata* (Vahl) Mast. essential oil. This wood-derived oil, containing high levels of cedrene (23%) and cedrenol (9.6%), demonstrated significant antimicrobial activity against several bacterial strains, including *E. faecalis*, *Escherichia coli* (Migula) Castellani & Chalmers, *Klebsiella pneumoniae* (Schroeter) Trevisan, and *Streptococcus* D [23].

The second most abundant compound, 2,1,3-benzothiadiazole (12.0%), has also been found in other plants, including *Armeria maritima* (Mill.) Willd. [24], *Lawsonia inermis* L. [25], and *Sambucus nigra* L. [26], as well as in natural products such as propolis [27]. It serves as a precursor for compounds with fungicidal, herbicidal, and antibacterial properties [28,29].

Concerning the other sesquiterpenoids and their derivatives, they show a widespread distribution across various plant families, in addition to featuring significant biological activities [30], as discussed below.

The presence of ledene has been reported in significant amounts, for instance, in the Australian tea tree oil (*Melaleuca alternifolia* (Maiden & Betche) Cheel) [31] and the *n*-hexane extract of *Barringtonia asiatica* (L.) Kurz seeds (33.6%) [32], with antimicrobial activity against *Staphylococcus aureus* Rosenbach and *E. coli*. That of ledene oxide has been documented —along with spathulenol, discussed below, and caryophyllene oxide— in *Salvia candidissima* Vahl. essential oil [33], with antifungal activity against *Rhizoctonia solani* J.G. Kühn and *Alternaria solani* Sorauer, and in *Shorea robusta* Gaertn F. methanolic extracts [34] together with alloaromadendrene oxide. In turn, high contents of isoledene have been reported in *Cryptomeria japonica* D. Don essential oils (12.4%), with antifungal activity against *Laetiporus sulphureus* (Bull.) Murrill, *Trametes versicolor* (L.) Lloyd, *R. solani*, *Colletotrichum gloeosporioides* (Penz.) Penz. & Sacc., *Fusarium solani* (Mart.) Sacc., and *Ganoderma australe* (Fr.) Pat. [35]. The non-polar twig extract of *Mundulea sericea* (Willd.) A.Chev. contained 29% of isoledene—and 18.2% of spathulenol—and showed activity against *S. aureus*, *E. coli*, *Bacillus subtilis* (Ehrenberg) Cohn, *P. aeruginosa*, and *Candida albicans* (C.P. Robin) Berkhout [36]. *Colocasia esculenta* (L.) Schott leaves’ ethanolic extract, rich in D-germacrene and isoledene, showed promising antibacterial activity against *Proteus vulgaris* Hauser, *E. coli*, *P. aeruginosa*, *S. aureus*, and *E. faecalis* as well as antifungal activity against *Cryptococcus neoformans* (San Felice) Vuill., *Aspergillus fumigatus* Fresenius, *Syncephalastrum racemosum* Cohn ex J. Schröt., and *C. albicans* [37].

The concurrent presence of ledol and spathulenol has been reported in several *Eryngium* species [38] and that of ledol, spathulenol, and alloaromadendrene in *Helichrysum petiolare* Hilliard & B.L.Burtt [39]. Ledol-rich *Tabauma gioi* A. Chev wood extract showed antifungal activity against *Trametes ochracea* (Pers.) Gilb. & Ryvarden [40], pure ledol synthesized from natural (+)-aromadendrene-III showed activity against *Cladosporium cucumerinum* Ellis & Arthur [41], while a ledol-enriched (12 %) essential oil showed antimicrobial activity against *S. aureus* (MIC = 50 µg·mL^−1^), *Bacillus cereus* Frankland & Frankland (MIC = 100 µg·mL^−1^), *Listeria monocytogenes* (Murray et al.) Pirie (MIC = 100 µg·mL^−1^), *E. faecalis* (MIC = 400 µg·mL^−1^), *Enterobacter aerogenes* Hormaeche & Edwards (MIC = 400 µg·mL^−1^), and *Campylobacter jejuni* (Jones et al.) Veron & Chatelain (MIC 30 µg·mL^−1^) [42].

Spathulenol is particularly widespread and has been found in *Annonaceae* (*Xylopia aromatica* Mart., *Xylopia emarginata* Mart. [43]), *Lamiaceae* (*Salvia sclarea* L. [44], *Salvia mirzayanii* Rech.f. & Esfand. [45], *Salvia yosgadensis* Freyn & Bornm. [46], and *Salvia rhytidea* Benth. [47]), and *Myrtaceae* (*Eugenia calycina* Cambess. [48], *Eugenia uniflora* L. [49], *Psidium guineense* Sw. [50], *Psidium guajava* L. [51], and *Psidium cattleianum* Afzel.ex Sabine [52]) as well as in other plants that do not belong to those families (e.g., *Barbacenia brasiliensis* Willd. [53]; *Cinnamomum osmophloeum* Kaneh. [54], *Croton arboreus* Millsp. [55], *Croton hirtus* L’Hér. [56], *Perilla frutescens* Britton [57], and *Zanthoxylum nitidum* DC. [58]). It has antiproliferative, anti-inflammatory, antinociceptive, and antimicrobial effects and acts as a mosquito repellent [50,59,60]. In terms of antimicrobial activity, spathulenol was effective against *C. neoformans* and *E. faecalis* (MIC = 200 µg·mL^−1^) [61]; *S. aureus* and *Staphylococcus epidermidis* (Winslow and Winslow) Evans (MIC= 1350 and 1500 µg·mL^−1^, respectively) [62]; *Micrococcus luteus* (Schroeter) Cohn, *B. subtilis*, and *Xanthomonas campestris* (Pammel) Dowson (MIC = 15 µg·mL^−1^) [63]; and *S. aureus*, *Proteus mirabilis* Hauser [44], and *Mycobacterium tuberculosis* (Zopf) Lehmann & Neumann (MIC = 231.9 µg·mL^−1^).

Alloaromadendrene, isolated from *Ambrosia peruviana* All., has been previously identified in *Duguetia glabriuscula* (R.E.Fr.) R.E.Fr. [15], *Ledum palustre* Michx. [64], *Monanthotaxis discolor* (Diels) Verdc. [65], *Prostanthera centralis* B.J.Conn [66], *Rhododendron tomentosum* (Stokes) Harmaja [67], and *Salvia bracteata* Banks & Sol. Alloaromadendrane has been reported to effectively inhibit *Cladosporium herbarum* (Pers.) Link growth [68] and has shown antimycobacterial activity against *M. tuberculosis* [69]. Aromadendrene, as alloaromadendrene, was found in *Eucaliptus* spp. [70], and related compounds such as aromadendrane-4-*β*,10*α*-diol, aromadendrane-4*α*,10*α*,diol, and 1-epimer-aromadendrane-4*β*,10*α*-diol from *Cinnamomum cassia* (L.) D.Don showed significant activity against *S. aureus* [71].

The presence of epiglobulol has been documented in *Achyranthes aspera* L. [72]; *Chamaemelum nobile* (L.) All.; *Matricaria chamomilla* L.; *Ligusticum striatum* DC. [73]; *Conyza dioscoridis* L. [74]; *Croton macrostachyus* Hochst. ex Delile [75]; *Eucalyptus nova-anglica* H.Deane & Maiden [76]; *Eucalyptus citriodora* Hook. [77]; *Jurinea auriculata* (DC.) N.Garcia, Herrando & Susanna [78]; *Ligusticum chuanxiong* S.H.Qiu, Y.Q.Zeng, K.Y.Pan, Y.C.Tang & J.M.Xu [79]; *Moringa oleifera* Lam. [80]; *Psidium guajava* L. [81]; *Salvia officinalis* L. [82]; and *Thottea barberi* Ding Hou [83]. Epiglobulol has been studied extensively for potential therapeutic applications for its anti-inflammatory and anticancer activities and is a natural insecticide [84]. As for its antimicrobial activity, *Croton tricolor* Klotzsch ex Baill. stem essential oil, in which epiglobulol is the main constituent (19%), showed promising antifungal activity against *Candida* spp. [85]

Cis-*α*-copaene-8-ol has been identified, for instance, in the essential oil from dried fruits of *Xylopia aethiopica* (Dunal) A.Rich. [86], in *Osyris laceolata* Hochst. & Steud. oil [87], in *Otostegia persica* (Burm.f.) Boiss. essential oil [88], and in *Houttuynia cordata* Thunb, with in silico molecular docking studies suggesting strong antibacterial activity [89]. This preconized activity was demonstrated against *B. subtilis* and *E. coli* in the case of essential oils from *Elsholtzia ciliata* (Thunb.) Hyl. [90].

Longiborneol is a constituent of various species including *Juniperus*, *Pinus*, *Cupressus*, *Dacrydium*, as well as *Cedrus deodara* (Lamb.) G.Don [91]. It has also been documented in *Mallotus tetracoccus* (Roxb.) Kurz leaves [92]; in the essential oils of *Cistus salviifolius* L., with antimicrobial activity against Gram-positive and Gram-negative bacteria [93]; in the leaf oil of *Cinnamomum chemungianum* Mohan & Henry, with moderate activity against certain strains of Gram-positive and Gram-negative bacteria [94]; and in the essential oil of *Drimys granadensis* L.fil., with activity against *L. monocytogenes* [95].

Aromadendrene has been detected at concentrations as high as 42.3% in *E. camaldulensis* fruit volatiles [16]. It was also the main constituent (9.1%) of the ether extracts of *Scapania verrucosa* Heeg., which showed antifungal activity against *C. albicans*, *C. neoformans*, *Trichophyton rubrum* (Castell.) Sabour., *A. fumigatus*, and *Pyricularia oryzae* Cavara [96].

The essential oil from leaves of *Withania adpressa* Coss. Ex, rich in longifolene (21.4%), showed antibacterial effects against *E. coli*, *K. pneumoniae*, *S. aureus*, and *Streptococcus pneumoniae* (Klein) Chester, while its antifungal efficacy was demonstrated against *C. albicans*, *Aspergillus flavus* Link, *Aspergillus niger* P.E.L. van Tieghem, and *Fusarium oxysporum* Schltdl. [97]. Its autoxidation products also showed antifungal activity against *T. versicolor*, *Lenzites betulinus* (L.) Fr., *Gloeophyllum trabeum* (Pers.) Murrill, *Trichoderma virens* (J.H. Mill., Giddens & A.A. Foster) Arx, and *Rhizopus oryzae* Went & Prins. Geerl. [98]. Its presence in agarwood essential oil —along with carvacrol— is thought to be responsible for the antifungal activity demonstrated in vitro and in vivo against agricultural and foodborne pathogens [99].

### 3.2. Structure–Activity Relationship: On the Biological Role of the Skeleton in Sesquiterpenes

As noted above, the characterization of the *S. chamaecyparissus* flower extract phytochemical profile revealed the presence of sesquiterpenoids with distinct skeletal types: compounds with a hydroazulene skeleton (consisting of a bicyclo[5.3.0]decane system, where a seven-membered ring is fused to a five-membered ring) and compounds with an aromadendrane-type.

The biological activity of compounds containing the aforementioned skeletons is influenced by several key structural features:(i)The degree of unsaturation affects both lipophilicity and molecular flexibility, which in turn influence bacterial membrane penetration. Studies with guaiane-type sesquiterpenes have shown that compounds with higher degrees of unsaturation typically exhibit enhanced antimicrobial activity due to improved membrane interactions [100,101];(ii)The presence and position of hydroxyl groups also significantly impact biological activity. Hydroxylated derivatives like ledol show enhanced antimicrobial properties through multiple mechanisms: increased water solubility facilitating cellular uptake, potential for hydrogen bonding with cellular targets, formation of reactive oxygen species in bacterial cells, and enhanced interaction with membrane phospholipids [102];(iii)The presence of a fused cyclopropane ring, as found in ledene and isoledene, introduces structural rigidity that can modify the three-dimensional conformation of the molecule, enhance binding specificity to cellular targets, increase molecular stability, and affect membrane permeability through changes in molecular shape [103,104];(iv)In alloaromadendrene, the presence of a *gem*-dimethylcyclopropyl unit rather than an isopropyl group confers antimicrobial activity, whereas its corresponding guaiane derivative shows in vitro cytotoxic activity against L-1210 cells [100]. Similarly, spathulenol has demonstrated biological activity, whereas its corresponding isopropyl derivative, nardol, has not shown any biological activity [105,106];(v)The stereochemistry of the ring junction between the five- and seven-membered rings plays a crucial role in determining the overall molecular shape and consequently its biological activity. The *cis*-fused hydroazulene system is more common in nature and typically shows higher biological activity compared to *trans*-fused analogs [103](vi)Additional functional groups and their spatial orientation can modulate activity: methyl groups increase lipophilicity, ketone groups can serve as hydrogen bond acceptors, and epoxide rings can react with cellular nucleophiles [107].

Looking at the key structural characteristics that influence bioactivity, a theoretical hierarchy of potential biological activity, from highest to lowest expected activity, may be proposed:

Ledene and alloaromadendrene oxide > (−)-spathulenol and ledol > 8-cedren-13-ol and *cis*-*α*-copaene-8-ol > epiglobulol and iso-ledene.

In comparing the basic structure of sesquiterpenoids containing hydroazulene and aromadendrene skeletons with that of the azulene-type (the unsaturated version of hydroazulen) or other sesquiterpenoids with the same biosynthetic origin (i.e., the cyclization of farnesyl pyrophosphate through different folding patterns and rearrangements) as germacrene and guaiane (Figure 6), the former should exhibit higher activity. This is evident when comparing them to the germacrene skeleton (a 10-carbon ring that can adopt pseudobicyclic conformations) [108], which has lower structural rigidity and shows only moderate antimicrobial activity. A similar effect occurs when comparing guaiane-type sesquiterpenoids (compounds with a decahydro-1,4-dimethyl-7-(1-methylethyl)azulene skeleton, a 5/7 system biosynthetically related to germacrene) from *Alisma orientale* (Sam.) Juz., *Enterospermum madagascariense* (Baill.) Homolle, *Amoora rohituka* (Roxb.) Wight & Arn., and *Curcuma phaeocaulis* Valeton [109,110,111,112], which exhibit lower activity. When the comparison is extended to sesquiterpenoids with an azulene skeleton, the antimicrobial activity of these is notably reduced due to aromaticity.

Briefly, based on structure–activity relationships, the aromadendrene and hydroazulene skeletons typically demonstrate higher antimicrobial activity than related parent compounds, particularly when functionalized with specific oxygenated groups (presence of either epoxide groups, secondary alcohols, and *gem*-dimethylcyclopropyl units) [100,102,107]. This enhanced activity can be attributed to their higher structural rigidity, better hydrophobic/hydrophilic balance, and consequently, improved ability to interact with cell membranes [103,104].

### 3.3. Comparisons of Efficacy

#### 3.3.1. Comparison Versus Previous Reports on the Antimicrobial Activity of *S. chamaecyparissus*

In this study, the hydromethanolic extract from *S. chamaecyparissus* flowers exhibited antifungal activity against *Neocosmospora* spp. with MICs of 1000–1500 µg/mL. Although there is extensive literature on the antimicrobial activity of *S. chamaecyparissus* (Appendix A), most previous studies [6,114,115,116,117,118] have focused on human pathogens and evaluated essential oils rather than hydromethanolic extracts, with reported MICs ranging from 0.625 to 1600 µg·mL^−1^ depending on the pathogen type, plant part used, and collection site. Studies on plant pathogens are scarce and have shown varying results: an aqueous extract demonstrated limited activity against *Fusarium oxysporum* f.sp. *lentis* W.L. Gordon (MIC > 20,000 µg/mL) [119], while a 30% commercial essential oil [120] showed differential mycelial growth inhibition against various plant pathogenic fungi, ranging from minimal activity against *Alternaria brassicae* (Berk.) Sacc. (10.65% inhibition) to substantial growth suppression of *Phytophthora parasitica* Dastur (72.02%), *Cladobotryum mycophilum* (Oudem.) W. Gams & Hooz (68.69%), *Fusarium oxysporum* Schltdl. (67.65%), and *Sclerotinia sclerotiorum* (Lib.) de Bary (54.50%). Notably, no activity was observed against *Botrytis cinerea* Pers. and *Pythium aphanidermatum* (Edson) Fitzp. at the tested concentration. Our findings extend the known spectrum of *S. chamaecyparissus* antifungal activity against other plant pathogens, representing the first report of its effectiveness against *Neocosmospora* species. The observed MIC values against *Neocosmospora* spp. are particularly promising when compared to the higher concentrations required for activity against other plant pathogenic fungi in previous studies.

#### 3.3.2. Comparison of Efficacy Against *Neocosmospora* spp. Versus Other Natural Compounds

Appendix A [24,121,122,123,124,125,126,127,128] presents a comparison of the reported efficacies of plant extracts and essential oils against the two *Neocosmospora* species studied. It is important to note that sensitivity may vary depending on the isolate, and results are expressed using different parameters (MIC values, percentage of mycelial growth inhibition (PMIG), and inhibition zones (IZ)), which makes direct comparisons challenging.

For *N. falciformis*, the flower extract of *S. chamaecyparissus* showed MIC values of 1500 µg·mL^−1^, positioning it in the intermediate-to-high range of efficacy among tested natural products. It was less effective than the hydromethanolic extract of *A. maritima* (MIC = 1000 µg·mL^−1^), the hydroethanolic extract of *Syzygium aromaticum* (L.) Merr. & L.M.Perry extract (MIC = 1000 µg·mL^−1^) [24,123], and the essential oils from *Cinnamomum aromaticum* Nees (MIC = 625 µg·mL^−1^) and *Cymbopogon flexuosus* (Nees ex Steud.) Will.Watson (MIC = 1250 µg·mL^−1^)) [122]. Nonetheless, it demonstrated better performance than several other treatments, which included *Citronella* spp. (requiring 20,000 µg·mL^−1^ for complete inhibition), *Melaleuca* spp. (achieving only 40.1% inhibition at 25,000 µg·mL^−1^) [121], and *Ocimum basilicum* L. essential oils (64.2–72% inhibition at 5000 µg·mL^−1^) [122]. Moreover, hydroethanolic extracts of *Hibiscus sabdariffa* L. and *Curcuma longa* L. showed much lower efficacy (MIC = 10,000 µg·mL^−1^), while *Cymbopogon citratus* (DC.) Stapf extract showed no significant activity (MIC > 10,000 µg·mL^−1^) [123].

Regarding *N. keratoplastica*, the *S. chamaecyparissus* flower extract (MIC = 1000 µg·mL^−1^) showed moderately good efficacy among tested products. Although it was less effective than *M. chamomilla* flower essential oil (MIC = 20 µg·mL^−1^) [128], it performed better than *A. maritima* flower extract (MIC = 1500 µg·mL^−1^) [24] and was comparable to *Origanum vulgare* L. subsp. *hirtum* essential oil (MIC = 800 µg·mL^−1^) [127]. Several other essential oils showed lower or no activity, including *Trachyspermum ammi* (L.) Sprague (no activity at the highest concentration tested) [124] and *Pogostemon cablin* (Blanco) Benth. (no activity at 500 µg·mL^−1^) [126]. *Kaempferia parviflora* Wall. ex Baker rhizome essential oil produced inhibition zones of 17–18 mm at 500 µg·mL^−1^ [125], but a direct comparison is not possible.

#### 3.3.3. Comparison of Efficacy Against *Neocosmospora* spp. Versus Conventional Fungicides

The MIC values obtained for *S. chamaecyparissus* L. flower extract (1500 and 1000 µg·mL^−1^ for *N. falciformis* and *N. keratoplastica*, respectively) can be compared with those previously reported by our group for commercial fungicides, as they were tested against the same fungal isolates under identical in vitro experimental conditions [24]. The currently prohibited dithiocarbamate Mancozeb demonstrated superior performance, achieving complete inhibition of both *Neocosmospora* species even at concentrations of 150 µg·mL^−1^, ten times lower than those required for the plant extract. The organophosphate fungicide Fosetyl-Al exhibited complete inhibition (100%) of both species at its recommended dose (2 mg·mL^−1^), which is higher than the extract’s MICs. The response to the strobilurin fungicide azoxystrobin was species-dependent: while *N. keratoplastica* was completely inhibited at the recommended dose (62.5 mg·mL^−1^), *N. falciformis* showed only partial inhibition (62.2%). Compared to these latter two conventional fungicides, the superior performance of the *S. chamaecyparissus* extract may be attributed to its particular composition, with a family of compounds potentially acting through different non-specific and specific mechanisms, as discussed in the recent review by Wiart et al. [129]. This characteristic could offer advantages for integrated disease management approaches, particularly in rotation schemes aimed at minimizing the risk of resistance development.

## 4. Materials and Methods

### 4.1. Reagents and Phytopathogens

Potato dextrose broth (PDB) and potato dextrose agar (PDA) were purchased from Becton Dickinson (Bergen County, NJ, USA). Ortiva^®^ (azoxystrobin 25%; Syngenta, Basel, Switzerland) and Fesil^®^ (fosetyl-Al 80; Bayer, Leverkusen, Germany) commercial fungicides were provided by the Plant Health and Certification Center (CSCV) of the Gobierno de Aragón.

*Neocosmospora falciformis* (MYC-1345) and *N. keratoplastica* (MYC-1250) isolates were supplied by the Mycology Lab of the Center for Research and Agrifood Technology of Aragón (CITA, Zaragoza, Spain) as living subcultures on PDA.

### 4.2. Plant Material and Extraction Protocol

Flowers of *S. chamaecyparissus* were collected during full bloom in June near the city of Huesca, NE Spain (42°09′13.6″ N 0°27′46.1″ W). A voucher specimen was authenticated by Prof. J. Ascaso and deposited at the herbarium of the Escuela Politécnica Superior de Huesca, University of Zaragoza. Samples from twenty-five specimens were combined to obtain representative composite samples, which were shade-dried, mechanically ground into powder, homogenized, and sieved through a 1 mm mesh.

For extraction, dried flower powder (34.2 g) was combined with 200 mL of a methanol/water solution (1:17 *v*/*v*). The mixture was heated for 30 min at 50 °C and sonicated using a probe-type ultrasonicator (model UIP1000 hdT; Hielscher Ultrasonics, Teltow, Germany). The mixture was subjected to ultrasonication at 20 kHz and 1000 W in alternating cycles: 10 to 15 min of sonication followed by 5 to 10 min rest periods to maintain the temperature between 30 and 60 °C. The extract was first decanted, then filtered through Whatman No. 1 paper, and subsequently centrifuged at 9000 rpm for 15 min. The resulting solution was freeze-dried to obtain the solid residue. The extraction yield was 18% (*w*/*w*) (6.4 g). For gas chromatography–mass spectrophotometry (GC-MS) analysis, the freeze-dried extract was dissolved in high-performance liquid chromatography (HPLC)-grade methanol to obtain a 5 mg·mL^−1^ solution, followed by filtration.

### 4.3. Physicochemical Characterization

The infrared vibrational spectra of both the flower heads (pre-extraction) and the freeze-dried extract were recorded using a Nicolet iS50 Fourier-transform infrared (FTIR) spectrometer (Thermo Scientific, Waltham, MA, USA) equipped with a diamond attenuated total reflection (ATR) system. Interferograms were generated by co-adding 64 scans across the 400–4000 cm^−1^ spectral range with a 1 cm^−1^ resolution.

A gas chromatography–mass spectrometry (GC-MS) instrument comprising a system with a model 7890A gas chromatograph coupled to a model 5975C quadrupole mass spectrometer (both from Agilent Technologies, Santa Clara, CA, USA) was used to elucidate the constituents of the flower hydromethanolic extract. This analysis was outsourced to the Research Support Services (STI) at Universidad de Alicante (Alicante, Spain). The chromatographic conditions consisted of an injection volume of 1 µL, an injector temperature of 280 °C in splitless mode, and an initial oven temperature of 60 °C held for 2 min, followed by a ramp of 10 °C·min^−1^ up to a final temperature of 300 °C, held for 15 min. An HP-5MS UI chromatographic column (30 m length, 0.250 mm diameter, and 0.25 µm film), also from Agilent Technologies, was employed for the separation of the compounds. The temperatures of the mass spectrometer’s electron impact source and quadrupole were 230 and 150 °C, respectively, with an ionization energy of 70 eV. Components were identified by comparing their mass spectra and retention times with those of authentic compounds and through computer matching with the database of the National Institute of Standards and Technology (NIST11). Test mixture 2 for apolar capillary columns according to Grob (Supelco 86501) and PFTBA tuning standards were used for equipment calibration.

### 4.4. In Vitro Antimicrobial Activity Assessment

Antifungal activity was evaluated using the agar dilution method following EUCAST standard procedures [130]. Aliquots of the stock solution were incorporated into the PDA medium to achieve final concentrations ranging from 62.5 to 1500 μg·mL^−1^. Mycelial plugs (5 mm diameter) were excised from the margin of 7-day-old PDA cultures of *N. falciformis* and *N. keratoplastica* and transferred to the center of the amended PDA plates. For each treatment and concentration, three replicate plates were prepared in duplicate. The control treatment consisted of PDA medium amended with the extraction solvent (methanol/water 1:17 *v*/*v*). All plates were incubated at 25 °C in darkness for seven days. Mycelial growth inhibition was calculated using the formula ((*d_c_* − *d_t_*)/*d_c_*) × 100, where *d_c_* and *d_t_* represent the mean colony diameters in control and treated plates, respectively. Effective concentrations (EC_50_ and EC_90_) were determined through PROBIT analysis using SPSS Statistics v. 25 (IBM, New York, NY, USA).

Since the Shapiro–Wilk [131] and Levene [132] tests indicated that the normality and homoscedasticity requirements were fulfilled, the mycelial growth inhibition results were subjected to one-way analysis of variance (ANOVA) and subsequent post hoc comparison of means through Tukey’s test at a significance level of *p* < 0.05, also conducted with IBM SPSS Statistics v. 25 software. In the case of in planta weight and root length results, normality and homoscedasticity requirements were only met for zucchini lengths (to which an ANOVA was applied). Kruskal–Wallis test [133] was used for zucchini weight and tomato weight and root length analysis, given that the distribution was not normal, but the groups were homoscedastic, followed by multiple pairwise comparisons using the Conover–Iman procedure.

### 4.5. In Vivo Tests on Horticultural Crops

Tomato cv. ‘Optima F1’ and zucchini cv. ‘Diamant F1’ plants for in vivo experiments were obtained from Agrodepa S.L. (Palencia, Spain) and Fronda Centros de Jardinería S.L. (Madrid, Spain), respectively.

The efficacy of the extract against *N. falciformis* was evaluated through artificial inoculation following modified versions of the methods described by González et al. [10] and Sánchez-Hernández et al. [134]. The fungus was grown in 250 mL flasks containing PDB for 3 days at 25 °C in the dark with constant shaking. Plant roots were first dipped in the treatment (either at MIC or at MIC × 2 concentration, i.e., 1500 and 3000 μg·mL^−1^) and then in a suspension of 5 × 10^6^ conidia·mL^−1^ for 2 min, after which they were transferred to the 4 cm × 4 cm plastic pots with sterilized peat substrate. Non-inoculated plants dipped in sterilized water were used as negative controls. Each treatment included twelve plants per replicate, with two independent replicates. Plants were maintained in a growth chamber at 25 °C with a 16/8 h photoperiod for 15 days, after which plant development, wilting, and yellowing were assessed.

## 5. Conclusions

The characterization and evaluation of bioactive compounds from *S. chamaecyparissus* flowers revealed their significant potential as natural fungicides. The extract’s composition was dominated by sesquiterpenoids (50.4% of total compounds), with distinctive structural features including hydroazulene and aromadendrane skeletons. Structure–activity relationship analysis supported that these skeletal types exhibit enhanced antimicrobial activity compared to analogous frameworks (such as azulene, germacrene, or guaiane), primarily due to their unique structural characteristics: higher molecular rigidity, optimal hydrophobic–hydrophilic balance, and specific functional group orientations.

The practical significance of this research lies in the extract’s demonstrated effectiveness against two emerging and polyphagous soil phytopathogens. The observed EC_90_ values (984.4 and 728.6 μg·mL^−1^ against *N. falciformis* and *N. keratoplastica*, respectively) and successful in planta protection at 3000 μg·mL^−1^ position this extract as a promising alternative to conventional fungicides. Its activity was comparable to fosetyl-Al and superior to azoxystrobin, suggesting its potential value in organic farming applications, where natural alternatives to synthetic fungicides are urgently needed.

Future research should focus on optimizing extraction methods, developing stable formulations, conducting field trials broadening the spectrum of plant hosts, utilizing various environmental conditions, and exploring potential applications against insect pests given the volatile nature of the active compounds.

The integration of natural product chemistry with agricultural applications demonstrated in this study contributes to the growing field of biorational pest management, offering a sustainable alternative to conventional synthetic fungicides while expanding the known applications of *S. chamaecyparissus*.

## Figures and Tables

**Figure 1 plants-14-00235-f001:**
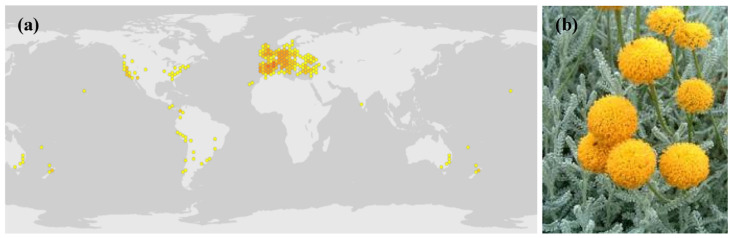
(**a**) *Santolina chamaecyparissus* L. worldwide distribution and (**b**) inflorescences. Source: *Santolina chamaecyparissus* L. in GBIF Secretariat (2023). GBIF Backbone Taxonomy. Checklist dataset https://doi.org/10.15468/39omei [2] accessed via GBIF.org on 11 December 2024.

**Figure 2 plants-14-00235-f002:**
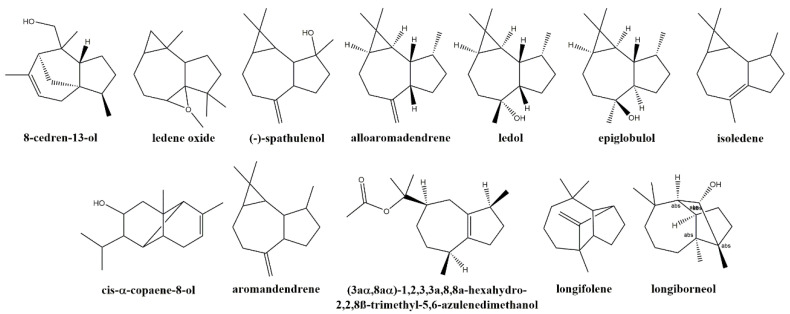
Main sesquiterpenoids identified in *S. chamaecyparissus* hydromethanolic flower extract, ordered from highest to lowest relative abundance (% of total content). Chemical structures were drawn using ChemDraw v.2023 (Revvity Signals Software, Waltham, MA, USA).

**Figure 3 plants-14-00235-f003:**
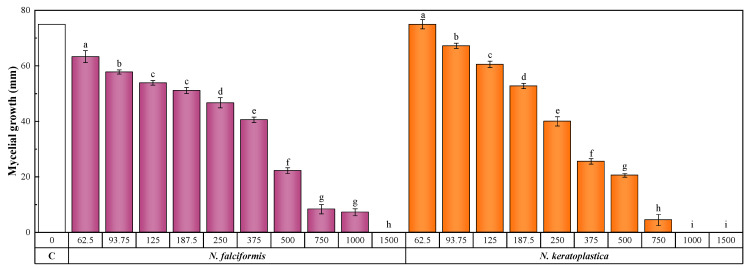
Inhibition of the growth of the mycelium of *N. falciformis* and *N. keratoplastica* in the in vitro tests performed with potato dextrose agar medium amended with different concentrations (in the 15.62–1500 µg·mL^−1^ range) of *S. chamaecyparissus* flower extract. C (white bar) represents the controls. The efficacies of the concentrations labeled with the same letters are not statistically different at *p* < 0.05. Standard deviations are represented by error bars.

**Figure 4 plants-14-00235-f004:**
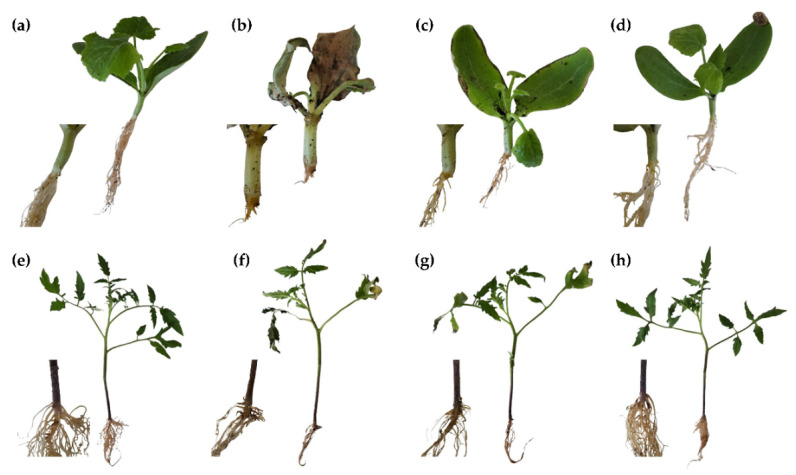
Symptoms of *N. falciformis* in (**a**–**d**) zucchini (*Cucurbita pepo* L.) cv. ‘Diamant F1’ and (**e**–**h**) tomato (*Solanum lycopersicum* L.) cv. ‘Optima F1’ plants 15 days after inoculation. (**a**,**e**) Negative controls with no inoculation or treatment, exhibiting healthy roots and foliage. (**b**,**f**) Positive controls (plants inoculated with *N. falciformis* and treated with bi-distilled water) showing severe symptoms, including leaf yellowing, wilting, and root rot; zucchini plants (**b**) also display stem base collapse. (**c**,**g**) Plants treated with *S. chamaecyparissus* flower extract at 1500 μg∙mL^−1^, which mitigates disease severity. (**d**,**h**) Plants treated with the *S. chamaecyparissus* flower extract at 3000 μg∙mL^−1^, showing no disease symptoms and appearing comparable to negative controls, with healthy roots and foliage.

**Figure 5 plants-14-00235-f005:**
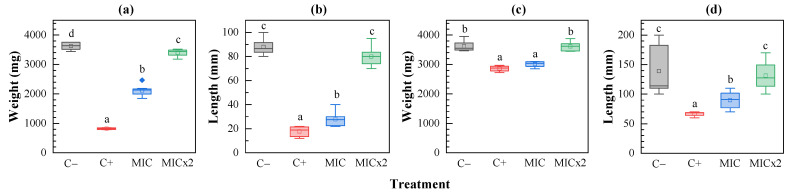
(**a**) Weight and (**b**) root length of zucchini plants; (**c**) weight and (**d**) root length of tomato plants. Different lowercase letters indicate significant differences at the 5% level of significance according to Tukey’s test (in the case of zucchini root length) or to Conover–Iman’s test (in the case of zucchini weight, tomato weight, and root length). C− and C+ stand for negative control and positive control, respectively. MIC and MIC × 2 represent the minimum inhibitory concentration (1500 µg·mL^−1^) and twice that concentration (3000 µg·mL^−1^), respectively.

**Figure 6 plants-14-00235-f006:**
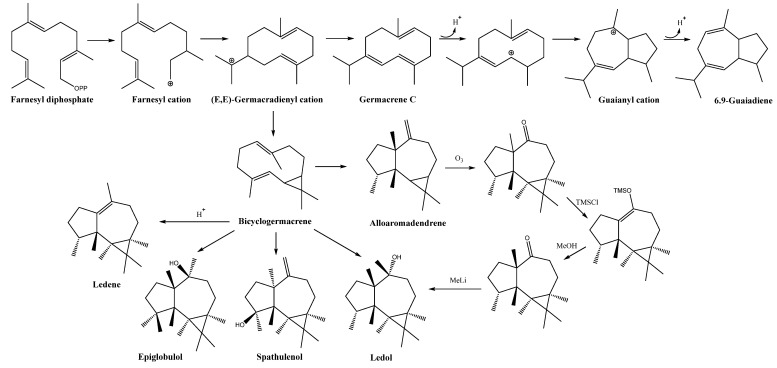
Types of primary cyclization of aromadrendane, bicyclogermacrene, and 6,9-guiadiene (adapted from [113]), along with the conversion of aromadendrene into ledol (adapted from [103]). Chemical structures were drawn using ChemDraw v.2023 (Revvity Signals Software, Waltham, MA, USA).

**Table 1 plants-14-00235-t001:** Main bands in the infrared spectrum of *Santolina chamaecyparissus* L. flowers before and after extraction. Wavenumber values are expressed in cm^−1^. Band assignments according to Socrates [13].

Flowers	Flower Extract	Assignment
3287	3292	O–H stretching (alcohols)
2919	2927	asymmetric aliphatic C–H stretching vibration (methylene)
1717	1694	C=O stretching of ketones and aldehydes
	1596	aromatic ring (C=C in-plane) stretching symmetric
1516	1515	C=C stretching (aromatic); C=O stretching
1372	1371	symmetric aliphatic C–H bending of CH_3_ groups (alkanes)
1239	1258	C–O stretch (in rings); out-of-phase C–C–O stretching
1147	1158	C–O–C group (O–bridge)
-	1115	C–O stretching in alcohols
1023	1031	C–O–H deformation in cellulose; C–O stretching (esters)
896	854	aromatic out-of-plane rings with two neighboring C–H groups
812	811	bending of C–H bonds in out-of-plane deformation in aromatics

## Data Availability

All data supporting the findings of this study are available within the paper and its Appendix A. Should any raw data files be needed in another format, they are available from the corresponding author upon reasonable request.

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
