# Peer review of "Bioactive Sesquiterpenoids from Santolina chamaecyparissus L. Flowers: Chemical Profiling and Antifungal Activity Against Neocosmospora Species"

_plants, 2025, doi:10.3390/plants14020235_

Round 1
Reviewer 1 Report
Comments and Suggestions for Authors
Title: Bioactive Sesquiterpenoids from Santolina chamaecyparissus L Flowers: Chemical Profiling and Antifungal Activity Against Neocosmospora Species
Summary: Sánchez-Hernández et al. investigated the phytochemical profile of Santolina chamaecyparissus flower against Neocosmospora spp. The bioactive compounds identified were predominately sesquiterpenoids and monoterpenes. The extract was evaluated against N. falciformis and N. keratoplastica in vivo and in vitro to achieve a higher inhibitory effect.
General comments: Antimicrobial activities of the phytochemical extract from Santolina chamaecyparissus have been previously investigated. Although the current work will expand the frontiers of this field, the results obtained here are limited. For instance, the authors did not include photos inhibitory assay performed. This is crucial in evaluating the results obtained. Moreover, since sesquiterpenoids and monoterpenes are members of the VOCs, it will be interesting to evaluate the extract against some insect pests as well. Furthermore, with respect to the wilting symptoms observed, authors can show a cross-section of the stem of the plants investigated, so it will be easier to link the wilting to the fungi infection. That notwithstanding, the methodology is robust, and the English language looks good, except for a few mistakes observed.
Specific comments
Line 19: Please replace “has been” by “was”
Sesquiterpenoids and monoterpenoids are members of the VOCs. Why did the authors not investigate their role against insect pests too?
Line 33: Please specify the type of inoculation. By spraying or injection?
I will encourage authors to add more background information on terpenes and narrow in on mono- and sesquiterpenoids and their antimicrobial/insecticidal functions in the introduction. Authors may find vital information here https://doi.org/10.3390/ijms22115710 Also briefly touch on the biosynthetic routes of mono- and sesquiterpenoids in the introduction.
Line 115: Figure 2 legend “……………listed by relative abundance.” What does this mean?
Line 119: Write MICs in full and abbreviate in subsequent lines.
Section 2.3. In vitro Antifungal Activity
Could authors show photos of the results (fungi plates) of the inhibitory assay performed?
Line 131: What does “Protection attained” mean?
Line 133-134: “…..controls (plants artificially inoculated and treated with bi-distilled water only) exhibited the typical symptoms for this pathogen, including yellowing and wilting of leaves, rotting….”
It will be interesting to observe the longitudinal section of the stem or root for wilting symptoms induced by the fungi spp.
Line 159: Please delete “On the”
Section 3.3.1. The title will be better off as “Antimicrobial Activity of S. chamaecyparissus” since the entire section has very little to do with a comparison of the present work and previous works.
The discussion is too lengthy. Authors should consider reducing summarizing it.
Comments on the Quality of English LanguageThe English Language looks good, except very few errors detected
Reviewer 2 Report
Comments and Suggestions for Authors
The work is excellent and highly relevant, particularly in the use of natural products as biorational solutions for crop protection, which is very compelling.
In introduction, the impact of Neocosmospora species on agriculture could be better addressed with numbers or statistics.
I recommend including the graphs obtained from the mass spectrometer and FTIR in the supplementary results, as they would be useful for confirming the results obtained.
For the FTIR and gas chromatography data, it would be beneficial to include references that corroborate the results presented in the tables and text.
Figure 4: Due to the nature of this image, I suggest increasing its size in the file to better highlight the details, especially those of the tomato plant.
Figure 4 Legend: The caption could also provide more detailed descriptions of the observed diseases, including the black spots on the roots and the brown and black marks on the leaves.
Line 166: A reference is missing.
Lines 169 and 171: The information within parentheses could be incorporated into the text for better readability.
Lines 180–183: The section discussing antimicrobial activity and its relationship with cedrene could be rewritten to clarify this connection.
Line 192: A reference is missing.
Lines 332, 336, 338: Some references are missing.
Lines 339–344: A reference is needed.
Line 368: This point could be explored further. What were the MIC concentrations of other extracts? Only one is mentioned in the text. What were the other concentrations used?
Line 500: Adjust the formatting and include the authors' names.

Reviewer 3 Report
Comments and Suggestions for Authors
The article provides a comprehensive overview of the phytochemical profiling and antimicrobial applications of Santolina chamaecyparissus L. (cotton-lavender). However, there are a few limitations that could be addressed:
1 The method for preparing the hydromethanolic extract is not detailed, which could lead to variability in the results. Standardization of the extraction process would ensure more reliable and reproducible data.
2 There are numerous issues regarding the non-standardized formatting of journal names in the references, necessitating a uniform writing style.
Round 2
Reviewer 1 Report
Comments and Suggestions for Authors
The authors have sufficiently addressed most of my comments and give satisfactory rebuttals to comments that were not addressed.
However, I encourage authors to add to their methods sections or legends the tool the used to draw the chemical structures.
Author Response
Q1. The authors have sufficiently addressed most of my comments and give satisfactory rebuttals to comments that were not addressed. However, I encourage authors to add to their methods sections or legends the tool the used to draw the chemical structures.
Response: We thank the reviewer for this suggestion. We have added information about the chemical structure drawing software (ChemDraw, Revvity Signals Software) in the figure legends for Figures 2 and 6, where chemical structures are presented.